



# Challenges and design choices for global weather and climate models based on machine learning

Peter D. Dueben[1] and Peter Bauer[1]

[1]European Centre for Medium-range Weather Forecasts, Shinfield Rd, Reading RG2 9AX

**Correspondence:** Peter Dueben (peter.dueben@ecmwf.int)

**Abstract.** Can models that are based on deep learning and trained on atmospheric data compete with weather and climate models that are based on physical principles and the basic equations of motion? This question has been asked often recently due to the boom of deep learning techniques. The question is valid given the huge amount of data that is available, the computational efficiency of deep learning techniques and the limitations of today's weather and climate models in particular with respect to resolution and complexity.

In this paper, the question will be discussed in the context of global weather forecasts. A toy-model for global weather predictions will be presented and used to identify challenges and fundamental design choices for a forecast system based on Neural Networks.

## 1 Introduction

In recent years, artificial intelligence and machine learning have become very important for hardware development in high performance computing (HPC) and have raised a large amount of public interest. Neural Networks (NNs) are tools from machine learning that are used successfully within many applications such as computer vision, speech recognition, and data filtering. If a sufficient amount of data is available, NNs can be trained to describe the evolution of non-linear processes. Due to the fundamentally application unaware character no complete understanding of the underlying process is necessary. Very complex NNs can be trained that use more than a billion trainable parameters and millions of datasets for training on HPC architecture; see for example Le (2013).

On the other hand, numerical weather forecasts are computationally expensive and forecast quality is reducing significantly already after a couple of days even in the best models available. Most processes of the Earth System are described by non-linear differential equations with non-linear interactions between Earth System components. Due to the complexity and size of the Earth System and the limited capacity of today's supercomputers, it is necessary to make approximations when weather prediction models are formulated and resolution is truncated in space and time. The use of limited resolution makes it necessary to parametrise processes that are not resolved explicitly within model simulations. To optimise parametrisation schemes a large number of parameters has to be tuned towards optimal model performance and the traceability of physical laws of the underlying process as well as the physical interpretation for each parameter is often lost during this exercise. Furthermore, to perform weather predictions, a huge amount of data needs to be processed and assimilated to create initial conditions. This





is a process that will again cause significant errors and uncertainties. Only a rather small fraction of all observations can be assimilated into state-of-the-art weather prediction models due to the large computational cost and simplified assumptions are required such as vanishing error correlation.

NNs have been used to post-process data from weather forecast models to optimise predictions; see for example Krasnopol-
sky and Lin (2012) or Rasp and Lerch (2018). NNs have also been used for radiation parametrisation in operational forecasts at ECMWF in the past (Chevallier et al., 1998, 2000; Krasnopolsky et al., 2005) as well as for the parametrisation of ocean physics (Krasnopolsky et al., 2002; Tolman et al., 2005) and convection (Krasnopolsky et al., 2013; Gentine et al., 2018). For parametrisation, NNs are either trained from observations or high-resolution model data with the ambition to provide better results compared to conventional parametrisation schemes or they are trained with input/output pairs of existing parametrisa-
tion schemes to emulate the behaviour and eventually replace the parametrisation scheme within forecasts. The latter is useful since NNs parametrisation schemes, that are based on very efficient HPC libraries such as TensorFlow (Ten, 2018) for which co-designed hardware exists, will in general be much more efficient compared to conventional parametrisation schemes that have a very large codebase that is difficult to optimize. Speed-up factors of up to $10^5$ have been observed; see Krasnopolsky and Fox-Rabinovitz (2006). It is possible, that NNs may become a standard tool to be used within the complex environment
of Earth System models to speed-up specific model components or to improve the representation of processes that cannot be represented adequately by physical equations. The use of NNs in the development of parametrisation schemes may also enable new approaches for representing model uncertainty in ensemble predictions.

Today many scientific groups around the world try to answer the more general question that is controversial: Can forecast models that are based on deep learning and trained on atmospheric data compete or even beat weather and climate models that
are based on physical knowledge and the basic equations of motion? Given the increasing number of meteorological observations that are available to train NNs, in particular since the beginning of the satellite era, the use of NNs may not be limited to parametrisation schemes and specific model components in the future. NNs may also become competitive with existing weather forecast models as a whole to perform actual weather predictions if observations of the past are used for training while observations of the present are used as input to generate forecasts. Global weather forecast models that solve three-dimensional,
non-linear equations may become obsolete. It can be assumed that more observations can be used for predictions in a weather forecast system based on NNs in comparison to predictions with dynamical models since data preprocessing and selection could be done by the networks and since higher resolution can be used for predictions since NNs can be expected to be much cheaper and easier to optimize for HPC in comparison to conventional models.

NNs have been used to generate local weather predictions (see for example Hall et al. (1999)) but it has yet to be shown
that forecasts based on deep learning can be competitive against operational weather forecast models in particular for global predictions. Answering this question is difficult since it requires to scale-up the training process of NNs to the level of complexity of a large supercomputing application to allow a fair comparison between the two approaches. In this paper we will make the first step and discuss the potential of NNs for global weather predictions. We base this discussion on tests with NNs that are used to represent the equation of the Lorenz'95 model – which is a low complexity model to test new approaches to
atmospheric modelling – as well as a NN toy-model for global weather predictions that is trained from atmospheric re-analysis



data. In both cases, no dynamical equations are used to update the model state. Results will be used to identify challenges and fundamental design choices for a forecast system based on NNs. Tests with Lorenz'95 serve as example for a system for which the basic equations are known while the exact equations are unknown for the toy-model for global weather predictions.

Our tests with Lorenz'95 and the toy-model for global weather forecasts are presented in section 2. Based on the results,
section 3 is discussing challenges and fundamental design choices for the development of forecast systems based on NNs. Section 4 presents the conclusions.

## 2    Results with Neural Networks

Section 2.1 will present results for initial tests with the Lorenz'95 low complexity model that serves as a test-bed for atmospheric dynamics. We will then develop a toy model for the global atmosphere that is used to calculate global weather forecasts
in Section 2.2.

NNs consist of neurons that resemble properties of neurons in the brain in terms of functionality and connectivity. In NNs, neurons are connected with each other and organised in layers. All networks that are used in the following are sequential NNs that use a linear stock of layers of neurons for which each neuron is connected to each neuron of the previous and subsequent layer, a so-called Multilayer Perceptron (MLP). Information travels from the inputs in the first layer to subsequent layers. The
outputs will leave the NN in the output layer. The layers between the input and output layer are called hidden layers. Each neuron is a weighted sum over all inputs (all neurons of the previous layer) plus a bias term ($\sum_{i=1}^{N} w_i n_i + b$; $w_i$ are weights, $b$ is the bias term). An activation function is applied to the accumulated value to represent non-linearity. During the training phase of the NN, the weights and biases within the network are optimized by reducing a loss function (mean absolute error for results in this paper). We use the Keras Python library (Chollet et al., 2015) to train and apply NNs for all results that
are presented in the following. We have tested several activation functions and optimizers and obtained the best results using hyperbolic tangents as activation function and a stochastic gradient descent optimizer. 20% of the training data is used for validation during training. All input and output data is normalized. All NNs that are used were trained for at least 200 training iterations that go through the entire data set during optimization, called epochs.

### 2.1    Initial tests with a toy model for atmospheric dynamics

We study the three-level Lorenz'95 model that was presented in Thornes et al. (2017). The model extends the original two-scaled Lorenz'95 model (Lorenz, 2006) by one more level such that it provides flexibility regarding tests at different resolution in a toy model for atmospheric dynamics. The model consists of three levels of model variables that can be assumed to be large scale "X", medium scale "Y" and small scale "Z". At each level, all degrees-of-freedom form a one-dimensional ring and the number of degrees-of-freedom is increasing by a factor of eight from one level to the next (8 for X, 64 for Y, 512 for Z).
Eight degrees-of-freedoms on a finer model level are coupled to one degree-of-freedom on the coarser level respectively. The degrees-of-freedom are described by the following differential equations:





$$\frac{dX_k}{dt} = X_{k-1}\left(X_{k+1} - X_{k-2}\right) - X_k + F - \frac{hc}{b}\sum_{j=1}^{J} Y_{j,k}, \tag{1}$$

$$\frac{dY_{j,k}}{dt} = -cbY_{j+1,k}\left(Y_{j+2,k} - Y_{j-1,k}\right) - cY_{j,k} + \frac{hc}{b}X_k - \frac{he}{d}\sum_{i=1}^{I} Z_{i,j,k}, \tag{2}$$

$$\frac{dZ_{i,j,k}}{dt} = edZ_{i-1,j,k}\left(Z_{i+1,j,k} - Z_{i-2,j,k}\right) - g_Z eZ_{i,j,k} + \frac{he}{d}Y_{j,k}. \tag{3}$$

The indices $i$, $j$ and $k$ range from 1 up to $I = 8$, $J = 8$, and $K = 8$ for the Z, Y and X tiers. $F$ is a large-scale forcing term
which determines the chaoticity of the model and is set to 20 in the simulations of this paper which results in fully chaotic
model dynamics. The remaining parameters allow tuning of the frequency and amplitude of oscillation as well as the coupling
between tiers and are set to $h = 1$, $c = b = e = d = 10$ and $g_Z = 1$ to obtain slow oscillations with large amplitude in the X tier
while the other two tiers oscillate more quickly at a lower amplitude. A fourth order Runge-Kutta method is used to integrate
the model in time using a timestep of 0.005 model time units (MTUs).

We consider a model simulation that is using all of the scales as a "truth". Similar to a weather prediction model that is
truncating spatial resolution, we can mimic limited resolution in the Lorenz'95 model by truncating the medium and small
scale degrees-of-freedoms.

We have trained several NNs to predict the tendency of the model ($\Delta X_k^n = X_k^{n+1} - X_k^n$) to update the state-vector in one
timestep. The NNs are used iteratively to make predictions for more than one timestep ($X_k^{m+1} = X_k^m + \Delta X_k^m$). To reduce the
error due to time discretization, the calculation of the right-hand-side tendency via the NN is coupled to a third order Adams-
Bashforth explicit time-stepping scheme. We use a first and second order scheme for the first two time-steps. The medium and
small scale variables ($Y_{j,k}$ and $Z_{i,j,k}$) are not represented in the NN, neither as inputs nor as outputs.

We use two different architectures for forecast. A "global" approach is using all eight $X_k$ variables as input to predict the
tendencies. On the other hand, a "local" approach is using $X_{k-2}$, $X_{k-1}$, $X_k$ and $X_{k+1}$ as input to predict a tendency for a
single variable $X_k$. For the local approach, the same NN is called eight times to update the entire state vector. The pairs of
training sets are separated by one model time unit in the truth run to generate data points that are sufficiently uncorrelated. We
use $2,000,000$ pairs of $X_k^{m+1}$ and $X_k^m$ to train the NNs. For "local" NNs the information for each variable $X_k$ of a training
set with a full state-vector is used as an independent training set.

We have performed tests with many different NN configurations with different layer width and number of layers. We achieved
the best results using four hidden layers between the input and output layer with 100 neurons per layer for the "global" setup
and two hidden layers with 20 neurons each for the "local" setup. These configurations were used in the following.

Figure 1 shows results for the two NNs. The trajectories that are calculated with the NNs show the typical dynamic of a
Lorenz'95 model and it is indeed possible to generate reasonable forecasts with the two models based on NNs. The error of
the local NN configuration is significantly lower compared to the global configuration. For comparison, we mimic a standard
forecast with limited resolution using a dynamic model that is based on equation (1) (see right plot in Figure 1). The medium
and small scale variables ($Y_{j,k}$ and $Z_{i,j,k}$) are not represented. No sophisticated parametrisation scheme is used and the coupling





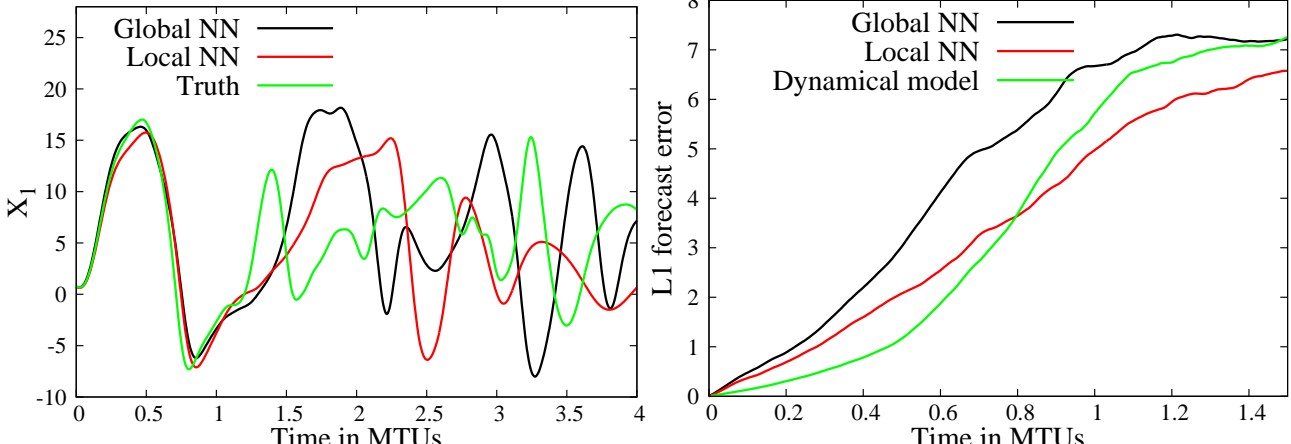

**Figure 1.** Left: Example trajectory for the "global" and "local" NN as well as the truth. Right: L1 error for forecasts with the "global" and "local" NN as well as the truncated dynamical forecast model.

terms for the degrees-of-freedoms that are not resolved are removed with no replacement (the last terms in equation (1)). The standard model is producing a lower forecast error compared to the global NN forecast systems and a lower forecast error for the local NN forecast system at the beginning of the forecast. However, the error for the local NN forecast system is lower towards the end of the forecast.

5 ## 2.2 A toy-model for global weather forecasts

We have developed a toy model to simulate the dynamics of the global atmosphere that can be used for global weather forecasts. We have focussed on the representation of geopotential height at 500 hPa (Z500) which is a standard field to analyse the quality of weather forecasts. Z500 was picked since the dependency on local conditions such as topography is limited (in contrast to fields such as surface pressure), since Z500 is not spotty with very strong local gradients (in contrast to fields such as humidity 10 or precipitation) and since most of the important global flow pattern – such as mid-latitude jets and a gradient between poles and equator – are visible.

### 2.2.1 General model setup

We use data of Z500 of the ERA5 re-analysis dataset (ERA, 2018) for training. Re-analysis data is the state of an atmospheric model with a continuous assimilation of observations to generate the best possible picture of the global atmosphere at a given 15 time. The advantage of the use of a re-analysis data set instead of observations for training is that data is available for each grid-point at each time-step and that the data is consistent over the entire data window. However, the use of analysis data restricts the use of the NN model to the forecast only. A conventional data-assimilation system is still necessary to generate initial conditions.



We use hourly data of Z500 from ERA5 for training and map the data to a longitude/latitude grid with 6 degrees resolution. We therefore consider global snapshots of Z500 with 60X31=1860 grid-points. 67,200 of these snapshots are available in total in the period between 1st January 2010 and 31st August 2017.

We have tested many different NN architectures and training configurations. Some changes to the setup had a significant influence on the quality of the toy-model. We will only present the most successful approaches in the following. We have played around with normalized fields that would use the anomaly field and remove the annual mean from the data. However, we found that the use of absolute field values achieved the best results. Similar to the NNs for the Lorenz model, all NNs are trained with Z500 data at the full hour $(n)$ to predict the tendency as the difference in Z500 $(\Delta Z500 = Z500^{n+1} - Z500^{n})$ one hour later $(n+1)$. This tendency is equivalent to the right-hand-side of a differential equation in time if the time-step is 1 hour. The NNs are used iteratively with a third order Adams-Bashforth explicit time-stepping scheme to make predictions for more than one hour into the future.

Time was used as additional input variable for all NNs. There is one coordinate that represents the daily cycle (growing linearly from midnight to midnight of the following day) and one coordinate to represent the annual cycle growing linearly from the beginning to the end of the year with correct representation of leap years.

In a second set of tests, we have included 2 meter temperature (2mT) as additional prognostic field in the NN forecast system. ERA5 data for 2mT was retrieved and processed in the same way as for Z500 and added as additional input and output parameters.

### 2.2.2 Global and local networks

As for the Lorenz case, we present results for a "global" and a "local" model configuration. "Global" networks are using all 60X31 grid-points plus the two time coordinates as input to calculate the 60X31 tendencies for each grid-point that are used to update the entire state-vector one hour into the future. This process is iterated to make predictions for longer lead times. "Local" networks are using a stencil of NXN points to calculate the tendency for a single grid-point in the centre of the stencil, with N being an odd integer number. This is similar to finite difference schemes in conventional models. We have trained NNs for different stencil sizes for the input fields (3X3, 5X5, 7X7, 9X9). For each stencil size, the same NN is used to update grid-points in the entire domain. However, to enable the NN to learn and represent local dynamics we have added the horizontal coordinates longitude and latitude as additional input variables. The approach of a local stencil is easy to realize but it generates a standard problem for all forecast models that are based on longitude/latitude grids: The pole requires special treatment.

To represent polar areas in "local" networks we have trained a special NN to update the north and south pole of the grid. The poles are represented by the first and last latitude band of the grid (60 grid-points for each pole but all of the points have the same value). The special pole NN was trained with the value of the pole plus the sixty variables of the next latitude band as input to output the tendency to update the pole. If we use a stencil of 5X5 grid-point inputs to update the bulk of the grid-points, we use the special pole NN to update the poles, a 3X3 stencil NN to update the latitude bands that are closest to the poles and the 5X5 stencil NN for all other points. To use a 9X9 stencil NN would require the use of a 7X7, a 5X5 and a 3X3 stencil NN as well as the special polar NN towards the two poles.





We have tested many different configurations of the NNs to identify the setups that produced the best results. For the results of the following section, we used four hidden layers of neurons that have the same width as the input layer (1862 neurons for the global and NXN+4 neurons for local configurations). For the NNs that use both Z500 and 2mT as input, the same configuration for neurons in hidden layers was used for the local configurations while the width of the hidden layers was increased to the

size of the new input vector for the global configuration and the number of hidden layers was reduced to two.

### 2.2.3  Results

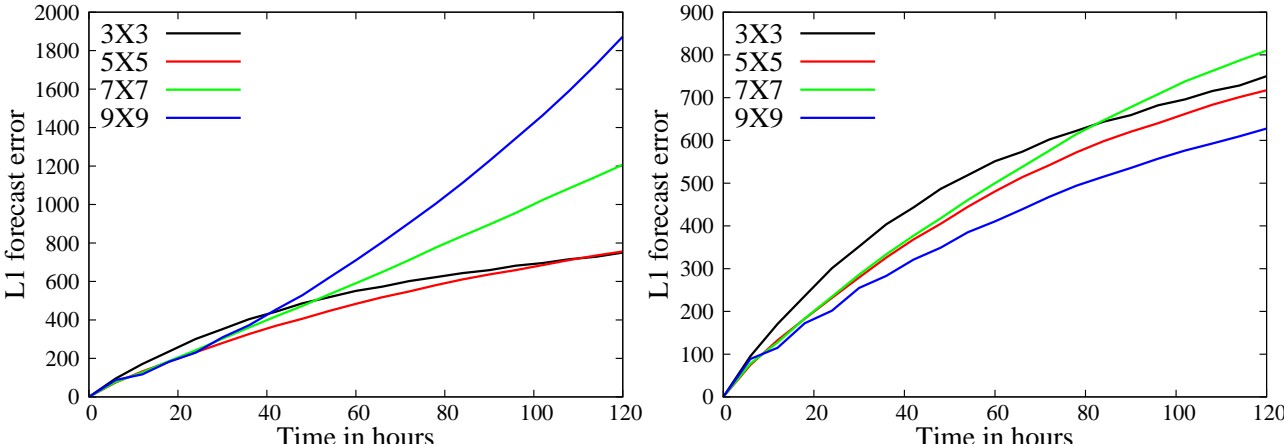

**Figure 2.** Globally integrated absolute forecast error plotted against time for "local" NNs with different stencil size. Left: NNs that use local networks with smaller stencil size as well as the special pole network towards the poles. Right: NNs that fix the fields close to the poles to stabilize simulations.

Figure 2 shows the global forecast error compared against the analysis that is used for operational forecasts at ECMWF for the "local" NN configuration and different stencil sizes. The forecast error was calculate as the average for ten forecasts distributed equally between March 2017 and February 2018. Some of the dates that have been used to calculate the forecast

error are also used when training the networks. However, this overlap is not a problem since a different analysis dataset has been used for initialisation. The forecast error is not very different for different stencil sizes at the beginning of the forecast. However, for the "local" networks that use the special treatment of the area around the pole, as discussed in the previous section, the forecast error is diverging for the 7X7 and the 9X9 configuration. When looking into the actual fields it became visible that this divergence is caused by chequerboard patterns developing close to the poles. We therefore started a second

set of simulations that used the original stencil size but kept all grid-points that could not be calculated with the largest stencil fixed throughout the forecast. These simulations remained stable for much longer lead times up to around two weeks and it is visible that the lines do not diverge in the same way towards the end of the forecast. The difference in forecast error between the two approaches is very small at the beginning of the forecasts.





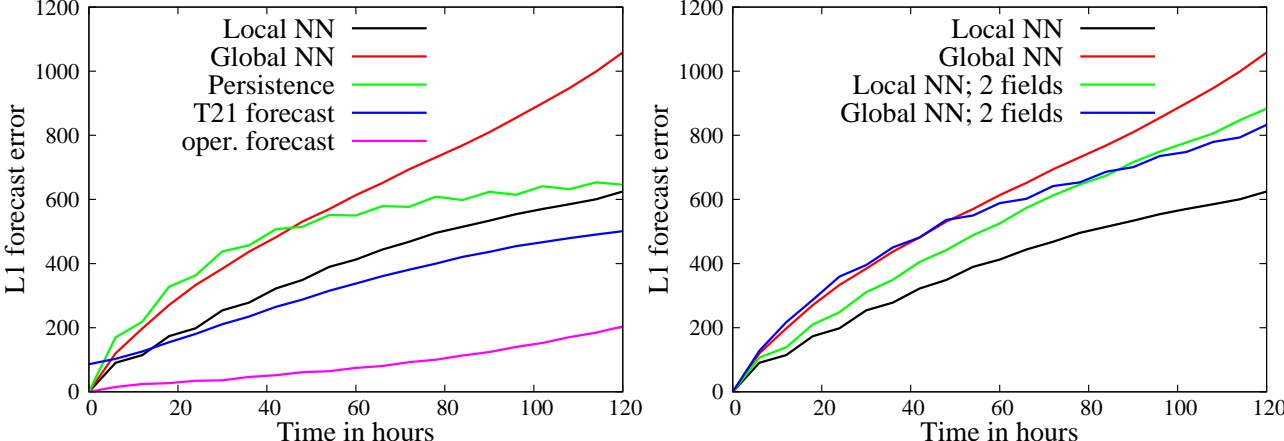

**Figure 3.** Left: Globally integrated absolute forecast error for the best "local" network (9X9 stencil), the "global" network, a persistence forecast, an IFS forecast at TL21 resolution and the operational weather forecast of ECMWF. The persistence forecast shows a 12 hourly fluctuation since Z500 has a weak 12 hourly cycle in the tropics due to atmospheric tides. Right: The same globally integrated absolute forecast error for the best local and global network as in the left plot plus the best results for local and global networks that use 2mT as additional prognostic field.

The left plot of Figure 3 shows the same global forecast error as Figure 2 when using different methods to generate the forecasts. We show results for forecasts with the Integrated Forecast System (IFS) at very coarse resolution (TL21 with 60 vertical levels) as well as operational forecasts. The TL21 resolution forecast is using a coarser horizontal resolution and model data was mapped to the six degree longitude/latitude grid to calculate the forecast error. Therefore, the initial error is

not zero for the TL21 forecast. We also show the persistence forecast error when assuming that the Z500 field will not change during the forecast window. The forecasts with "local" NNs beat TL21 forecasts at the beginning of the forecast as well as the persistence forecasts during the full five day period. The "global" network shows little benefit compared to the persistence forecast.

The right plot of Figure 3 shows the global forecast error for Z500 when 2mT was added as prognostic field in the forecast

system. The quality of forecasts for Z500 did not improve in comparison to the forecast system based on Z500 only.

Figure 4 shows the Z500 fields for a one day forecast with a single field as well as the change of Z500 during the first day of the forecast with the "local" and "global" NN configuration. The analysis that is used for operational forecasts at ECMWF is also presented. The Z500 fields that are generated with the NNs look healthy and reasonable after one day with no obvious problems in the solution. Generated videos that visualise the development of the field in time (attached as supplemented

material) also look very realistic for the first couple of days. The difference between the observed tendencies and the "local" network are small during the first forecast day. The "global" network seems to underestimate the magnitude of changes.



**Figure 4.** Top: Z500 from analysis at 1st March 2017 that is used as initial conditions for forecasts. Middle: Z500 for the analysis at 2nd March for reference (left) as well as the "local" network and the "global" network configurations one day into the forecast (middle and right). Bottom: Difference between Z500 for the analysis at 1st and 2nd March (left) as well as the difference between initialization and after 24 hours for the "local" and the "global" network configuration (middle and right). The "local" network is using a 9X9 stencil and fixed polar regions.

## 3   Challenges and fundamental design choices for forecast systems based on NNs

Results of the previous section show that it may indeed be possible to generate global weather predictions based on NNs for short-range prediction and at this rather course spatial resolution. Whether a NN prediction system will ever be competitive with state-of-the-art weather prediction models remains an open question. It would certainly require a serious level of complexity
5   with as many (or more) degrees-of-freedom as conventional models. To develop such a system a couple of important decisions





need to be made regarding the structure of the networks and the shape of the training data. We can only guess how the optimal configuration of a NN prediction system may look like. This section will identify and discuss some of the important challenges and design choices.

### 3.1 Local or global networks and time-stepping schemes

Should forecasts with NNs be iterative or should they be trained for specific lead times? Should global forecasts be performed by a single NN that takes as many data inputs as possible or would it be better to use many NNs that are coupled together?

State-of-the-art weather forecast models use up to a billion variables to represent the state of the Earth System. Atmosphere and ocean show chaotic behaviour with scale-interactions that result in exponential error growth. Global weather forecast models have skill for several forecast days into the future and can be used for seasonal predictions. This requires models to be able to represent complex interactions between weather features such as convection in the tropics that generates gravity waves to influence jet position in the mid-latitudes. To enable forecast systems based on NNs to be competitive in global weather predictions several days into the future would require to (1) represent all relevant scales and features all over the globe, (2) allow scale-interactions, and (3) be able to represent chains of complex interactions between weather features.

Similar to the use of explicit or implicit time-stepping methods for conventional models there seem to be two approaches to achieve (1)-(3) with NNs. Either all information is connected, such as in implicit methods, to allow large time-steps, or the connectivity of networks is local and time-steps are short, such as in explicit time-stepping schemes. The sheer size of the Earth will make it difficult to connect all information in "global" networks and if global all-to-all communication across many compute nodes of an HPC system would be required, NNs would loose much of their performance benefit in comparison to conventional models on modern HPC facilities. High-resolution global networks could certainly not be dense, meaning that all neurons between different layers are connected as in MLPs. The global networks at very coarse resolution that were used in the paper already use a very large number of tunable parameters in comparison to NNs used in other disciplines due to their dense character. To reduce the amount of trainable parameters with no loss in scale interactions will be important.

On the other hand, "local" networks would need to be applied in iterative ways to allow interactions between scales and features in predictions for longer lead times. It can be argued that the lead time needs to be adjusted to the size of the local stencil of inputs. If "local" networks are used, there is another decision to be made: Shall the same network be used for all grid-points or shall different networks be used for different locations around the globe. In this paper, we are using the same NN for all grid-points. However, an approach that would use a different network for different locations could be realised using convolutional networks that combine stencil information locally within one of the layers but do not propagate information throughout the entire grid. This would allow to update a large part of the domain with a single network and to exploit standard network configurations for image processing, at least if structured grids are used. However, it would require significant work to make sure that boundary conditions are represented correctly. This approach was therefore not tested in this paper.

Can we assume that NNs can use timesteps that are as long or longer than timesteps of implicit timestepping schemes in conventional models? This should be possible in principle since a complex network with global inter-node communication could, in an extreme case, resemble an implicit timestep of a conventional model one-to-one. However, the amount of training



data and network complexity that is necessary will quickly become prohibitive if interactions between more and more features and longer chains of causality shall be represented.

Results with our toy-model suggest that:

- It is fundamentally possible to generate global weather forecasts when connecting iterative time-stepping schemes with NNs. This conclusion is entirely based on very coarse resolution data and derived from assessing the predictive skill for a variable that varies much slower in space and time than, for example, precipitation. However, none of the network configurations that we tested allowed stable simulations for integrations of more than two weeks into the future and forecast skill quickly deteriorates after a couple of days. If networks are applied iteratively, it will be important to satisfy fundamental conservation properties and to stabilize simulations. This is non-trivial and will require attention, in particular if it is the aim to generate climate predictions. It is also likely that model biases – that are difficult to distinguish from forecast variance – will perturb predictions of a NN forecast system.

- It will be difficult to train networks to make predictions with long lead time with a single time-step. When we tried to train NNs to make prediction with longer time-steps than 1 hour, results were degraded significantly. A linear reduction of the timestep will most likely be required if resolution is increased, similar to CFL requirements for conventional models.

- Results are much better for the "local" approach when compared to the "global" approach for both the Lorenz'95 model and the toy model for global predictions. We decided to use the same network to update grid-points all over the domain instead of using special networks for each grid-point. This approach appears more promising since the resulting network is more likely to be consistent with physical laws since it is trained for many different physical situations. Local information can be represented even if the network is used in the entire domain when adding spatial coordinates to the inputs. If a customized network is used for each grid-point or if a convolutional network is used to connect local stencils, the amount of data that is available for training is reduced by the number of grid-points and the representation of extreme events within the training data will often be questionable, in particular in a changing climate.

### 3.2 Understanding of the physical system

How far can we go using a "black box"? NNs allow to solve non-linear systems as a black-box with no knowledge of the actual physical system. There is no reason why weather forecasts could not be generated with such a black box. However, the work on the toy-model clearly indicated that a physical understanding is still important to improve the NN architecture and training to perform weather predictions.

One example is the time-stepping scheme that was used. In a first approach, we have tried to generate daily or six hourly forecasts with a single step with only very limited success. When thinking about possible problems, we remembered that an explicit time-stepping scheme would require much shorter steps at the given level of model resolution. Due to our knowledge on the level of numerical complexity that is required to allow weather and climate models to run at longer timesteps, we



concluded that it will be much easier for the NN to learn the correct dynamics for shorter timesteps and switched the dataset from ERA Interim (Dee et al., 2011) to ERA5 to allow training with hourly data. Results improved significantly.

A second example relates to the pre-processing of input and output data. Meteorological data comes in very different shapes. For example: while specific humidity seems to be an easy quantity since it is confined between 0 and 1, very small values can
still be very important and values can change by orders of magnitude between the stratosphere and the troposphere within a single vertical level; precipitation can have very significant outliers in extreme events; fields such as geopotential height or temperature show global gradients and local features which makes it difficult to judge whether gradients or absolute fields are important; fields such as surface pressure depend heavily on local features such as topography. If these fields are used as input or output for NNs to predict weather, physical knowledge of the properties of the fields and their connectivity will be essential
to design the optimal data pre-processing and the optimal network architecture. It may be possible to heal shortcomings of data pre-processing and network configurations with a brute force increase in data volume and training time as long as limits in numerical precision do not remove information from data. However, success will be limited in particular since many trial and error tests at large computational cost will be necessary to find the optimal network configuration for complex configurations that have the ambition to compete with conventional weather forecast models.

At the given state of software and science it is very difficult for domain experts to evaluate and understand connectivity of data in NNs to be in a good position to improve predictions. Networks are treated as a black box. The case of the Lorenz'95 model is a good example. If sub-grid-scale parameters are ignored for the moment, the differential equations and correct connectivity between the prognostic variables are known. It should be possible to design and train NNs that resemble the exact behaviour of the equation and to compete with the dynamic model. Since parameters are adjusted to the data, networks should
also be able to automatically incorporate parameterisation schemes for sub-grid-scale variables and to beat dynamic models that do not have parametrisation schemes. However, it was unclear how to project our knowledge into the design of the NN and the training procedure. As a result we did not manage to obtain results that were close to optimal. However, we do not claim that we have tested all possible options. While it is likely that even small changes in the network setup may have improved results significantly, the parameter space that needs to be explored to find the optimal configuration has many dimensions
(activation function, #layers, #neurons, #epochs, #data sets, data normalization, connectivity between neurons,...). This makes the search for the optimal configuration cumbersome. Tests with 2mT as additional field in the global toy model also showed that it is difficult to relate different model fields to each other and that it is not sufficient to just add input information to improve predictions. A physical understanding of connectivity and signal propagation is much easier in conventional models that allow to assign a physical meaning to all parameters. To develop tools and approaches for domain scientists to understand
and improve connectivity within complex NNs will be essential.

## 3.3 Data

What data should be used for training and how should it be pre-processed? How can observational data be assimilated to generate initial conditions for forecasts?





A vast amount of meteorological data is available that could be used for training of forecast models. The data handling system of ECMWF provides access to over 210 petabyte of primary data and the data archive of ECMWF grows by about 233 terabyte per day (ECM, 2018). However, it is questionable how much of this data can actually be used for training and the numbers shrink down substantially dependent on choices regarding network architecture and the selection of training data.

If unfiltered observational data is used as input for networks, biases between different observation systems need to be addressed and networks need to be robust against missing or faulty input data. In a first step, it would probably be necessary to use data from a small number of sources that provide a continuous spatial coverage rather than a moving data window. Even if, for example, all satellite data could be used for training such data would only be available for a couple of decades.

It is also questionable how much data can actually be used given that the climate of the Earth is constantly changing and

in particular given the rapid changes due to anthropogenic climate change. This will also decrease the time-frame for which data can be used for training since we cannot expect a model that is trained as a black box to provide reliable predictions if the underlying climate state has changed and if events that have never happened in the training data start to happen in the real world, such as an ice-free Arctic during summer. Conventional models show significant biases in long-term simulations. These biases will also be a problem for models based on NN and may change the local "climate" within a couple of days of

simulations and push the network to weather regimes that were not covered by the training data.

For a model that is based on physical principles, it can be assumed that changes in the general circulation or the frequency and shape of extreme events due to climate change can be represented correctly, at least within limits. "Local" networks that are trained to represent the dynamics at all grid-points, as in the toy-model, will be better suited for simulations in a changing climate compared to networks that are trained for local conditions at each grid point since they have been trained in the context

of different local climate.

Existing weather and climate models could be used to generate a shear unlimited amount of training data to train a forecast system based on NNs also for a changing climate. Re-analysis data, that was used for the toy-model, could be used as well. However, the quality of the existing models and the assimilation system would limit the quality of predictions with the NN forecast system, reducing the advantage of the deep learning approach. Tests with the toy model clearly indicate that it will

require data with very high temporal resolution to develop a NN forecast system that can run at high resolution, similar to the length of time-steps in conventional models, that will not be available in standard re-analysis datasets or for standard long-term model integrations and require a very large data system.

The results with the toy-model suggest that it is difficult enough to develop forecast models if consistent data with full spatial coverage is used for training. This suggests to start investigations towards a NN forecast system with re-analysis or model data

which is less noisy and more consistent when compared to real-world observations. If it is possible to replicate the dynamics of the model in sufficient details, more difficult tasks such as the use of real observations as input, can be addressed.



# 4   Conclusions

We have developed a toy-model based on NNs to generate global weather forecasts. The toy-model is not using any dynamical equation of motion. The model is based on a six degree longitude/latitude grid and is representing Z500 as the only model field. We show that it is indeed possible to make predictions into the future that are better than a simple persistence forecast and
competitive with forecasts of very coarse-resolution (TL21) atmosphere models of similar complexity at least for short lead times. We did not intend to build a model that can be competitive with operational weather forecast models. However, we use the toy-model to identify challenges and to indicate fundamental design choices that may lead to optimal results for forecast systems based on NNs.

While the use of deep learning techniques is often discussed as being a silver bullet to represent non-linear systems, it has yet
to be shown whether weather forecast systems based on NNs can be competitive compared to conventional models in particular for global forecasts, for longer time ranges than one day, and across the wide range of physical parameters that are provided by numerical models with physical consistency. The experience with the toy-model suggests that there will be no free lunch. While NNs can, in principle, be used as a black box, the development of a weather forecast system will require domain knowledge about the Earth System. Close collaborations between computer scientists and meteorologists will be essential even if petabytes
of training data and exascale supercomputers are available. A deep understanding on how to use physical knowledge of the Earth System and the connectivity between degrees of freedoms to improve the development of Network architectures and Network training and how to preserve conservation properties will be required.

The development of a NN forecast model will face similar challenges when compared to the development of conventional models such as the complexity of the Earth System with non-linear interactions between model components, scale interactions,
exponential growth of errors in initial conditions, numerical instabilities and the discrete representation of model fields on the sphere (leading to the pole problem in our toy-model), the treatment of conservation properties, model biases in long-term simulations, errors in observations and insufficient data coverage of observations. On the one hand, it is likely that NN models could make better use of future computing hardware and use more observations and higher resolution. On the other hand, it is also likely that it will be difficult to stabilize long-term integrations (in particular in a changing climate) and to represent
complex interactions between model features in long simulations since it will be hard to improve physical consistency within networks. It therefore seems obvious to focus on short-term forecasts and potentially also regional predictions in a first approach to develop forecast systems based on NNs. It will be difficult for NN models to compete with conventional models in long-range weather forecasts and climate predictions.

*Code availability.*   The code that was used to generate the result of this paper was attached to the publication as supplemented material.

*Data availability.*   Only standard ERA5 re-analysis data (see ERA (2018)) has been used for training.



*Acknowledgements.* Peter D. Düben gratefully acknowledges funding from the Royal Society for his University Research Fellowship and the ESIWACE project. The ESIWACE project has received funding from the European Union's Horizon 2020 research and innovation programme under grant agreement No 675191. Many thanks to Christoph Angerer, John Griffith, Peter Watson and Nils Wedi for very helpful comments and advise.



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
