# Peer review of "Challenges and design choices for global weather and climate models based on machine learning"

_Geoscientific Model Development, 2018_

## Referee Comment (RC1) · Anonymous Referee #1 · 17 Jul 2018

General comments

Overall, this is a nice piece of pilot work and a stimulating discussion about the potential for machine learning to change how we make numerical weather forecasts, and in particular the potential for an 'equation-free' approach that doesn't even use discretized equations of fluid motion. In particular, it highlights the important of an intelligent human design for such an approach to be competitive.

It is interesting (but reassuring given the structure of the underlying equations of atmospheric motion) that a 'local' approach works better for both problems considered than a 'global' approach. The machine learning global forecast approach uses limited inputs

(Z500 and optionally T2m, at 6 degree lat/lon resolution) but performs comparably to a version of the current ECMWF operational model with similar resolution (TL21), though not nearly as well as the full-resolution operational model. If you used a polar filter to remove high zonal wavenumbers in the NN-predicted Z500 near the poles, could you avoid the ad hoc approach of fixing Z500 at polar gridpoints to maintain the stability of your NN? Specific comments

Page 2 Line 7: There is some relevant new literature on machine learning used for atmospheric physical parameterization, e. g.: Schneider et al. (2017). Earth system modeling 2.0: A blueprint for models that learn from observations and targeted high-resolution simulations. GRL, 44, 12,396–12,417. https://doi.org/10.1002/2017GL076101 Brenowitz N.D., and C.S. Bretherton, 2018: Prognostic validation of a neural network unified physics parameterization, Geophys. Res. Lett., 45, https://doi.org/10.1029/2018GL078510.

The Schneider et al. paper is particularly relevant to the discussion here, as it uses a similar Lorenz 1995-type toy problem .

Page 2 Line 9: Add comma before 'or'

Page 2 Line 14: Remove comma after 'possible'.

Page 3 Line 30: Should be 'degrees of freedom'

---

## Referee Comment (RC2) · Anonymous Referee #2 · 24 Jul 2018

The authors show that it will be a challenge to develop neural nets that will beat NWP models. I agree. They also state that it will be difficult to beat long-range (e.g. seasonal) forecasts. However, given the dominance of model error (biases not least) on these longer timescales, here I am much less convinced. What evidence is there that neural nets for ENSO, for example, could not at least equal in skill those based on coupled models?

The authors correctly state that one of the reasons for developing neural nets for parametrisations is to make them much cheaper computationally. Here the neural nets can be trained on the parametrisations themselves. Unfortunately the Lorenz 95 sys-

tem is too simple to really test this idea - it would be trivial to develop a neural net for the Y and Z variables which would be competitive with a simple cubic fit or whatever. It would be good if the authors could speculate on how to test this latter idea in a way which was challenging for the neural nets on the one hand, but without going to full GCMs on the other.

I recommend this paper for publication subject to the authors addressing these two points.

---

## Author Comment (AC1) · 8 Aug 2018

Many thanks to both reviewers for the constructive comments that helped to improve the paper. We will list the comments of the reviewers and our responses in the following.
**Reviewer 1**

General comments

*Overall, this is a nice piece of pilot work and a stimulating discussion about the potential for machine learning to change how we make numerical weather forecasts, and in particular the potential for an "equation-free" approach that doesn't even use discretized equations of fluid motion. In particular, it highlights the important of an intelligent human design for such an approach to be competitive.*

*It is interesting (but reassuring given the structure of the underlying equations of atmospheric motion) that a "local" approach works better for both problems considered than a "global" approach. The machine learning global forecast approach uses limited inputs (Z500 and optionally T2m, at 6 degree lat/lon resolution) but performs comparably to a version of the current ECMWF operational model with similar resolution (TL21), though not nearly as well as the full-resolution operational model. If you used a polar filter to remove high zonal wavenumbers in the NN-predicted Z500 near the poles, could you avoid the ad hoc approach of fixing Z500 at polar gridpoints to maintain the stability of your NN?*

**Response:**

Many thanks for the positive review and the very useful feedback. We agree, that a polar filter or a relaxation against a reference state may remove the spurious behaviour at the poles. In fact, many of the mechanisms that are used to stabilise conventional grid-point models are likely to work in a very similar way for a Neural Network forecast system. Targeted customisation of the network architectures around the poles may also help to remove the spurious behaviour. However, this would require trial-and-error testing and would not be based on any theoretic understanding.

We feel that the application of a polar filter is beyond the scope of this paper. However, we are now mentioning the possible fix via a filter when discussing the problems around the poles on page 5:

"It is likely that the instabilities that were found near the poles could be removed using a polar filter or a relaxation of the model fields against a reference solution. These methods have been used to stabilize conventional grid-point models near the poles in the past. It is also possible that the pole problem can be solved via a change of the network architecutre of the NNs that are used in the vicinity of the poles. However, a more detailed investigation of this problem is beyond the scope of this paper."

Specific comments

1. *Page 2 Line 7: There is some relevant new literature on machine learning used for atmospheric physical parameterization, e. g.:*
   *Schneider et al. (2017). Earth system modeling 2.0: A blueprint for models that learn from observations and targeted high-resolution simulations. GRL, 44, 12,396-12,417.*
   *Brenowitz N.D., and C.S. Bretherton, 2018: Prognostic validation of a neural network unified physics parameterization, Geophys. Res. Lett.*
   *The Schneider et al. paper is particularly relevant to the discussion here, as it uses a similar Lorenz 1995-type toy problem.*

   **Response:**

   Yes, many thanks for the heads-up. We have updated the publication list with four new entries:

   Brenowitz, N. D. and Bretherton, C. S.: Prognostic Validation of a Neural Network

[Figure]

Unified Physics Parameterization, Geophysical Research Letters, 2018.

O'Gorman, P. A. and Dwyer, J. G.: Using machine learning to parameterize moist convection: potential for modeling of climate, climate change and extreme events, ArXiv e-prints, 2018.

Rasp, S., Pritchard, M. S., and Gentine, P.: Deep learning to represent sub-grid processes in climate models, ArXiv e-prints, 2018.

Schneider, T., Lan, S., Stuart, A., and Teixeira, J.: Earth System Modeling 2.0: A Blueprint for Models That Learn From Observations and Targeted High-Resolution Simulations, Geophysical Research Letters, 2018.

We have also changed the relevant text in the introduction:

"NNs have been used to post-process data from weather forecast models to optimise predictions; see for example Krasnopolsky and Lin (2012) or Rasp and Lerch (2018). NNs have also been used for radiation parametrisation in operational forecasts at ECMWF in the past (Chevallier et al., 1998, 2000; Krasnopolsky et al., 2005) as well as for the parametrisation of ocean physics (Krasnopolsky et al., 2002; Tolman et al., 2005) and convection (Krasnopolsky et al., 2013). Recently, the representation of atmospheric sub-grid processes using techniques from machine learning was investigated in more detail with promising results using both NNs (Brenowitz and Bretherton, 2018; Gentine et al., 2018; Rasp et al., 2018) and random forest decision trees (O'Gorman and Dwyer, 2018). For parametrisation, NNs can be trained from observations or high-resolution model data, for example from high resolution simulations of the same model, simulations that use superparametrisation, or large-eddy simulations, with the ambition to provide better results compared to conventional parametrisation schemes (see for example Schneider et al. (2018)). NN parametrisation schemes can also be trained with input/output pairs of existing

parametrisation schemes to emulate the behaviour and eventually replace the parametrisation scheme within forecasts. The latter is useful since NNs parametrisation schemes, that are based on very efficient HPC libraries such as TensorFlow (Ten, 2018) for which co-designed hardware exists, will in general be much more efficient compared to conventional parametrisation schemes that have a very large codebase that is difficult to optimize. Speed-up factors of up to $10^5$ have been observed; see Krasnopolsky and Fox-Rabinovitz (2006). It is possible that NNs may become a standard tool to be used within the complex environment of Earth System models to speed-up specific model components or to improve the representation of processes that cannot be represented adequately by physical equations. The use of NNs in the development of parametrisation schemes may also enable new approaches for representing model uncertainty in ensemble predictions."

However, we do not feel that the results with the Lorenz model in Schneider et al. are particularly relevant for this paper. Schneider et al. is using the Lorenz model in the rather different context of model parameter estimation.

2. *Page 2 Line 9: Add comma before "or". Page 2 Line 14: Remove comma after "possible". Page 3 Line 30: Should be "degrees of freedom".*

**Response:**

Many thanks. All of this has been changed.

**Reviewer 2**

*The authors show that it will be a challenge to develop neural nets that will beat NWP models. I agree. They also state that it will be difficult to beat long-range (e.g. seasonal) forecasts. However, given the dominance of model error (biases not least) on these longer timescales, here I am much less convinced. What evidence is there that neural nets for ENSO, for example, could not at least equal in skill those based on coupled models?*

*The authors correctly state that one of the reasons for developing neural nets for parametrisations is to make them much cheaper computationally. Here the neural nets can be trained on the parametrisations themselves. Unfortunately the Lorenz 95 system is too simple to really test this idea - it would be trivial to develop a neural net for the Y and Z variables which would be competitive with a simple cubic fit or whatever. It would be good if the authors could speculate on how to test this latter idea in a way which was challenging for the neural nets on the one hand, but without going to full GCMs on the other.*

*I recommend this paper for publication subject to the authors addressing these two points.*

**Response:**

Many thanks for the positive review and the very useful feedback.

Regarding point 1: Yes, it is very true that we should not cover all applications of NNs in our statements about predictability of longer timescales at the end of the conclusions. We are convinced that NN forecast models that learn the equation of the atmospheric state and rely on high complexity and short timesteps (as discussed in the paper) will have problems with biases similar to the problems observed in conventional models.

This is for example discussed on page 13:

"These biases will also be a problem for models based on NNs and may change the local "climate" within a couple of days of simulations and push the network to weather regimes that were not covered by the training data."

However, we agree that the results of this paper cannot be used to judge whether NNs that are trained to predict large-scale features such as ENSO may be able to beat conventional models. We have therefore changed the discussion of the paper in the final paragraph of the conclusion:

"The development of a NN forecast model that is based on a model grid as discussed in this paper will face similar challenges when compared to the development of conventional models such as the complexity of the Earth System with non-linear interactions between model components, scale interactions, exponential growth of errors in initial conditions, numerical instabilities and the discrete representation of model fields on the sphere (leading to the pole problem in our toy-model), the treatment of conservation properties, model biases in long-term simulations, errors in observations and insufficient data coverage of observations. On the one hand, it is likely that NN models could make better use of future computing hardware and use more observations and higher resolution. On the other hand, it is also likely that it will be difficult to stabilize long-term integrations (in particular in a changing climate) and to represent complex interactions between model features in long simulations since it will be hard to improve physical consistency within networks. For NN forecast systems that try to describe the evolution of the atmosphere based on gridpoints after learning the right hand side of the equations of motion from data, it therefore seems obvious to focus on short-term forecasts and potentially also regional predictions in

a first approach. It will be difficult for these models to compete with conventional models in long-range weather forecasts and climate predictions. However, the same statement may not be true for other applications of NNs that do not propagate the full atmosphere to make long-term forecasts but rather focus on predictions of large-scale flow patterns such as ENSO and weather regimes."

Regarding point 2: Yes, to train a NN to represent the impact of sub-grid-scale variables that are truncated (Y and Z) in a "coarse model" that is only using X variables is almost too simple. This was in fact a task for John Griffith (a Master student at Bristol University who was co-supervised by Peter Dueben). The results for forecast errors with a NN and a conventional parametrisation scheme can be seen in Figure 1. The model configuration that used a NN to learn the sub-grid-scale parametrisation term is showing a lower forecast error. The next step could be to learn the terms for sub-grid-scale closure for models of medium complexity such as shallow water or QG when comparing simulations at low and high resolution. However, it can be argued that the replacement of parametrisation schemes has already left the level of "toy-models" with (1) the successful application of Neural Networks to replace the radiation scheme of IFS by Chevallier et al. or (2) the replacement of atmospheric sub-grid processes by a NN that was trained from super-parametrisation runs and successfully applied in free-running aqua-planet simulations by Rasp et al. (both papers are cited in this paper). We are currently also trying to repeat the work of Chevallier at ECMWF to replace the existing radiation scheme of IFS at full complexity for weather forecasts by Neural Networks to speed-up model integrations. Therefore, and also since the paper is not focussing on parametrisations but rather the learning of the entire equations of motion, we would like to avoid a more detailed discussion on how to use the Lorenz model to learn more about the replacement of parameterisations within this paper.

However, there are indeed still very interesting questions regarding the use of NNs for parametrisation schemes that can be studied in the Lorenz model. One of the

questions is how to represent sub-grid-scale variability when using NNs. Figure 2 shows more results of John Griffith on the use of concrete dropout and Generative Adversarial networks to represent this variability in a NN parametrisation scheme. The results show, that it is indeed possible to represent some of the variability when using neural networks to develop parametrisation schemes. However, the AR1 stochastic parametrisation scheme is still performing best for the representation of sub-grid-scale variability and further work would be needed to achieve competetive results with NNs.

[Figure]

Interactive
comment

**Fig. 1.** Forecast error in the Lorenz system using a conventional parametersation scheme based on a polynomial fit as well as a parametrisation scheme that was using a NN to learn the sub-grid-scale parametris

[Figure]

[Figure]

**Fig. 2.** Spread-error plots for simulations with the Lorenz model that use a stochastic AR1 parametrisation scheme (left), a NN that is using the concrete dropout technique (middle) as well as a NN that is usi